# Factors Associated with Large Cup-to-Disc Ratio and Blindness in the Primary Open-Angle African American Glaucoma Genetics (POAAGG) Study

**DOI:** 10.3390/genes14091809

**Published:** 2023-09-16

**Authors:** Anusha Mamidipaka, Isabel Di Rosa, Roy Lee, Yan Zhu, Yineng Chen, Rebecca Salowe, Victoria Addis, Prithvi Sankar, Ebenezer Daniel, Gui-Shuang Ying, Joan M. O’Brien

**Affiliations:** Department of Ophthalmology, Scheie Eye Institute, University of Pennsylvania, Philadelphia, PA 19104, USA; anusha.mamidipaka@pennmedicine.upenn.edu (A.M.); isabel.dirosa@pennmedicine.upenn.edu (I.D.R.); roy.lee@pennmedicine.upenn.edu (R.L.); yan.zhu1@pennmedicine.upenn.edu (Y.Z.); yineng.chen@pennmedicine.upenn.edu (Y.C.); rebecca.salowe@pennmedicine.upenn.edu (R.S.); victoria.addis@pennmedicine.upenn.edu (V.A.); prithvi.sankar@pennmedicine.upenn.edu (P.S.); ebdaniel@pennmedicine.upenn.edu (E.D.); gs.ying@pennmedicine.upenn.edu (G.-S.Y.)

**Keywords:** primary open-angle glaucoma POAG, cup-to-disc ratio, vision loss, African American, neurodegeneration

## Abstract

Background/Aims: Primary open-angle glaucoma (POAG) disproportionately affects individuals of African ancestry. In these patients’ eyes, a large cup-to-disc ratio (LCDR > 0.90) suggests greater retinal ganglion cell loss, though these patients often display varied visual ability. This study investigated the prevalence and risk factors associated with LCDR in African ancestry individuals with POAG and explored the differences between blind (>20/200) and not blind (≤20/200) LCDR eyes. Methods: A case–control methodology was used to investigate the demographic, optic disc, and genetic risk factors of subjects in the Primary Open-Angle African American Glaucoma Genetics Study. Risk factors were analyzed using univariable and multivariable logistic regression models with inter-eye correlation adjusted using generalized estimating equations. Results: Out of 5605 eyes with POAG, 1440 eyes (25.7%) had LCDR. In the multivariable analysis, LCDR was associated with previous glaucoma surgery (OR = 1.72), increased intraocular pressure (OR = 1.04), decreased mean deviation (OR = 1.08), increased pattern standard deviation (OR = 1.06), thinner retinal nerve fiber layer (OR = 1.05), nasalization of vessels (OR = 2.67), bayonetting of vessels (OR = 1.98), visible pores in the lamina cribrosa (OR = 1.68), and a bean-shaped cup (OR = 2.11). Of LCDR eyes, 30.1% were classified as blind (≤20/200). In the multivariable analysis, the statistically significant risk factors of blindness in LCDR eyes were previous glaucoma surgery (OR = 1.72), increased intraocular pressure (OR = 1.05), decreased mean deviation (OR = 1.04), and decreased pattern standard deviation (OR = 0.90). Conclusions: These findings underscore the importance of close monitoring of intraocular pressure and visual function in African ancestry POAG patients, particularly those with LCDR, to preserve visual function.

## 1. Introduction

Primary open-angle glaucoma (POAG), the most common form of glaucoma, is an optic nerve degeneration that is the leading cause of irreversible blindness worldwide [1]. POAG disproportionately affects African ancestry individuals; these individuals are over five times more likely to be diagnosed with POAG and up to 15 times more likely to experience glaucomatous vision loss from the disease than individuals of European descent [2]. Cup-to-disc ratio (CDR), defined as the ratio of the optic cup diameter to the optic disc diameter, is a commonly used measurement to assess and track the progression of glaucomatous damage to the optic nerve. In terms of its diagnostic power, CDR has been considered one of “the most valuable optic disc variables for early detection of glaucomatous nerve damage” [3]. Cupping often occurs in POAG due to progressive neuroretinal rim narrowing [4] and mechanical stress on the lamina cribrosa caused by elevated intraocular pressure (IOP) [5]. Vertical CDR (VCDR), which is measured on the vertical meridian, has been shown to have a strong genetic correlation with POAG susceptibility [6]. A large VCDR (LCDR) suggests greater ganglion cell loss due to displacement of the lamina cribrosa and loss of axon fibers [4,7,8]. This axonal compromise, in addition to the mechanical failure of connective tissue in the lamina cribrosa, peripapillary sclera, and scleral canal wall, is a central pathophysiology that underlies glaucomatous damage [5].

Current studies on LCDR are limited, despite its evaluative importance in POAG and other ophthalmic diseases. Existing studies in European and Asian ancestry populations have found that risk factors for LCDR include higher IOP [6], greater lamina cribrosa depth [9], older age, and male sex [10]. While several studies have identified a significant correlation between LCDR and visual field loss in POAG subjects [11,12,13,14,15], others have found no such correlation, suggesting that intrasubject asymmetry in central corneal thickness (CCT) explains the varying levels of visual field damage in glaucoma patients [16]. Another study found no significant difference in CDR among POAG groups with visual field progression and without progression [17]. Interestingly, this suggests that, despite having axonal loss and displacement of the lamina cribrosa due to cupping, at least a subset of patients with LCDR maintain normal visual outcomes that do not progress to blindness. However, there is a lack of literature regarding why some eyes with LCDR maintain normal vision while others progress to blindness, especially in the overaffected African ancestry population.

In this paper, we examined the risk factors for presentation with LCDR and subsequent progression to visual impairment or blindness in a large cohort of African ancestry individuals. These individuals were recruited from the Primary Open-Angle African American Glaucoma Genetics (POAAGG) study. The goal of this paper was to understand the demographic, optic disc, phenotypic, and genetic characteristics that predispose these individuals to extreme cupping, and, among those with LCDR, to determine the risk factors for blindness. A better understanding of these risk factors can help to inform decisions on treatment escalation among individuals presenting with LCDR.

## 2. Materials and Methods

### 2.1. Study Design and Population

A case–control methodology is used in the POAAGG study. The POAAGG study cohort (details described elsewhere) consists of 10,255 individuals of African descent recruited from the greater Philadelphia area [18]. Each enrolled subject was classified by a glaucoma specialist or ophthalmologist as a case, control, or suspect based on detailed criteria. Each subject signed an informed consent form and provided a DNA sample. The degree of each subject’s African ancestry was quantified using the fastSTRUCTURE program, described elsewhere [19]. The Declaration of Helsinki was followed, and research was carried out with Institutional Review Board clearance from the University of Pennsylvania.

### 2.2. Demographic Characteristics

Based on previously recognized criteria, a glaucoma expert categorized participants as POAG cases, POAG suspicions, or controls [18]. Participants who also had an eye injury history, non-glaucomatous optic disc neuropathy, inflammatory eye disorders, Grave’s disease with ocular symptoms, or any type of glaucoma other than POAG were excluded. At enrollment, Clinical Research Coordinators performed standardized interviews to gather data on systemic illness, behavior, and demographics. Demographic data included age, gender, self-described race, usage of cigarettes and alcohol, hypertension, and diabetes. Electronic medical records were evaluated to determine previous glaucoma surgery history to supplement the onsite examination data.

### 2.3. Quantitative Phenotypic Characteristics

At enrollment, ocular phenotypic data for subjects were collected from both eyes, including IOP, CDR, CCT, visual acuity (VA), retinal nerve fiber layer (RNFL) thickness, mean deviation (MD), and pattern standard deviation (PSD). Eyes with very low vision (less than 20/200) were assigned the following logMAR values: 2.2 for counting fingers (CF), 2.3 for hand motion (HM), 2.4 for light perception (LP), and 2.5 for no light perception (NLP).

### 2.4. Qualitative Phenotypic Characteristics

Baseline 30-degree color stereo disc photographs of POAAGG subjects were collected using the Topcon TRC 50EX retinal camera (Topcon Corp. of America, Paramus, NJ, USA) and were uploaded onto a secure server at the Scheie Image Reading Center. Three non-physician graders were trained by glaucoma specialists to grade digital stereo color images of the optic disc using a stereo viewer (Screen-Vu stereoscope) [20]. Using a standardized grading form, two readers independently graded quantitative and qualitative features of the optic disc, and differences were adjudicated by the ophthalmologist director of the Reading Center. The qualitative optic disc characteristics measured were the presence of beta peripapillary atrophy, stereoscopically identified disc tilt, disc hemorrhage, arteriole narrowing, venule narrowing, baring of the lamina cribrosa, nasalization of the vessels, cilioretinal vessels, grey crescent, disc pallor, and visible pores in the lamina cribrosa. Additionally, graders characterized the disc shape as round, oval, or other and the cup shape as conical, cylindrical, or bean pot.

### 2.5. Definition of LCDR

This study included eyes from POAG cases. Eyes with severe optic nerve cupping (VCDR: 0.90–1) were defined as having an LCDR. The maximum VCDR value was determined during clinical eye examinations by glaucoma specialists. A typical retinal fundus photograph of the patients with LCDRs and those of the healthy control is provided in Figure 1.

### 2.6. Genotype

We previously conducted a genome-wide association study (GWAS) for POAG in 11,275 individuals of African ancestry (6003 cases, 5272 controls), including the African Descent and Glaucoma Evaluation Study (*n* = 1999) [21], the Genetics of Glaucoma in People of African Descent (GGLAD) consortium (*n* = 2952) [22], and the POAAGG study (*n* = 6324). We discovered 46 risk loci associated with POAG at genome-wide significance in a discovery mega-analysis, with replication analyses, trait colocalization analyses, and functional studies implicating three likely causal loci: rs1666698 mapping to *DBF4P2*, rs34957764 mapping to *ROCK1P1,* and rs11824032 mapping to *ARHGEF12.* We selected these three SNPs and tested their association with LCDR and with blindness in this study. Each variant was stratified into three groups based on the number of risk alleles (0, 1, and 2) present. Generalized estimating equations accounted for the inter-eye correlation.

### 2.7. Statistical Analysis

We performed a descriptive analysis for risk factors (demographic, phenotypic, and optic disc) using mean (SD) for continuous factors and using proportion for categorical factors. We performed the risk factor analysis for LCDR and blindness using univariable and multivariable logistic regression models. Generalized estimating equations accounted for the inter-eye correlation. The initial univariate analysis included all risk factors with a *p*-value < 0.2, and backward variable selection was used to reach the final multivariate analysis that retained only risk factors with *p* < 0.05. All statistical analyses were performed in R version 4.03 and SAS v9.4 (SAS Institute Inc., Cary, NC, USA).

## 3. Results

A total of 5605 eyes from POAAGG cases met eligibility criteria for this study, with 1440 eyes (25.7%) meeting the criteria for LCDR. Univariate analyses of demographic and clinical (Appendix A), optic disc (Appendix A), and phenotypic (Appendix A) characteristics of LCDR eyes were conducted.

A multivariate analysis of risk factors for the development of LCDRs was also conducted (Table 1). Out of 5605 eyes, 1890 eyes were included in the final multivariable model due to missing data in the univariate variables: previous glaucoma surgery (*n* = 350), IOP (*n* = 11), MD (*n* = 1392), PSD (*n* = 1384), RNFL (*n* = 1335), VA (*n* = 674), shape of cup (*n* = 2322), vessel bayonetting (*n* = 2310), nasalization of the vessels (*n* = 2203), and visible pores in lamina cribrosa (*n* = 2306).

LCDR was significantly associated with previous glaucoma surgery (odds ratio (OR) 1.72, 95% CI 1.23–2.42, *p* = 0.002), 1 mmHg increase in IOP (OR 1.04, 95% CI 1.02–1.06, *p* = 0.0006), vessel bayonetting (OR 1.98, 95% CI 1.44–2.74, *p* < 0.0001), vessel nasalization (OR 2.67, 95% CI 1.92–3.73, *p* < 0.0001), visible pores in the lamina cribrosa (OR 1.68, 95% CI 1.18–2.40, *p* < 0.0001), bean = pot-shaped cup (OR 2.11, 95% CI 1.35–3.31, *p* < 0.0001), 1 unit decrease in MD (OR 1.08, 95% CI 1.06–1.11, *p* < 0.0001), 1 unit increase in PSD (OR 1.06, 95% CI 1.01–1.12, *p* = 0.02), worse VA (1 logMAR unit increase, OR 1.88, 95% CI 1.32–2.65, *p* = 0.0004), and 1 μm decrease in RNFL thickness (OR 1.05, 95% CI 1.04–1.07, *p* < 0.0001).

The univariable analysis of genetic variants and LCDR in glaucoma cases (Table 2) did not unveil significant associations between the studied SNPs and LCDR.

To further investigate why a subset of eyes with LCDR progress to blindness while others maintain normal VA, we conducted additional analyses to compare the two groups. Among the 1440 LCDR eyes, 186 eyes were excluded due to missing VA values. Of the 1254 eyes analyzed, 448 had normal VA (with worst VA ≤ 20/40), 429 had impaired VA (VA between 20/40–20/200), and 377 were blind (VA > 20/200) (Figure 2). To simplify the univariate and multivariate analyses of risk factors for blindness, the eyes of the normal VA and impaired VA groups were combined into one group (*n* = 877) and compared to the blind group (*n* = 377).

A univariate analysis of demographic characteristics (Appendix A), optic disc (Appendix A), and phenotypic (Appendix A) characteristics of LCDR eyes predicting blindness was conducted. A multivariate analysis of risk factors for the development of blindness among eyes with LCDR was also conducted. Out of 1254 eyes, 440 eyes were excluded due to missing data in previous glaucoma surgery (*n* = 79), MD (*n* = 366), and PSD (*n* = 366). In this multivariate analysis, previous glaucoma surgery (OR 1.72, 95% CI 1.17–2.53, *p* = 0.0062), a 1 mmHg increase in IOP (OR 1.05, 95% CI 1.03–1.06, *p* < 0.0001), 1 unit decrease in MD (OR 1.04, 95% CI 1.02–1.07, *p* < 0.0001), and 1 unit increase in PSD (OR 0.90, 95% CI 0.85–0.95, *p* = 0.002) were significantly associated with blindness (Table 3).

The univariable genetic analysis (Table 4) did not reveal significant associations between the studied genetic variants and blindness among LCDR eyes.

## 4. Discussion

This study aimed to characterize risk factors associated with the development of LCDR and to assess predictors of blindness among African ancestry individuals with POAG. Among the African ancestry cohort, we found 26.5% of eyes met the criteria for LCDR (CDR > 0.9). LCDR was significantly correlated with worse overall visual field presentations, worse VA, higher IOP, previous glaucoma surgery, thinner RNFL, presence of visible pores in the lamina cribrosa, and vessel bayonetting and nasalization. In total, 377 (30%) of the 1254 LCDR eyes were blind (VA < 20/200). Among LCDR eyes, blindness was associated with previous glaucoma surgery, higher IOP, and worse visual field presentation characterized by higher MD.

The prevalence and risk factors associated with LCDR differ among individuals of African, Asian, and European ancestries. According to studies, non-Hispanic White individuals with glaucoma had smaller CDRs, thicker CCT, and less severe visual field defects compared to other ethnicities [23,24]. Conversely, on average, African Americans have a larger CDR compared to White populations [25,26]. The African Descent and Glaucoma Evaluation Study found that African-descent individuals have greater optic disc areas than European-descent individuals [27]. In fact, studies have found optic discs to be significantly smaller in White and Hispanic populations compared to Asian and African-ancestry ones [28,29]. In studies on Asian populations, the risk factors for LCDR were found to be higher IOP, lower diastolic and higher systolic blood pressure, higher AST/ALT ratio, longer axial length, lower BMI, older age, male sex, and previous cataract surgery [30,31]. Notably, higher IOP emerged as the most significant determinant of a LCDR, similar to our findings in an African ancestry population [31].

In our study, eyes with LCDR were more likely to experience elevated IOP, which is a major risk factor for POAG [32,33]. Studies have consistently found a significant correlation between higher IOP and LCDR, an indication of more severe glaucomatous damage [6,30,34]. Furthermore, research has shown that both IOP and VCDR have significant heritability and are genetically correlated with each other, as well as with the risk of POAG [6,34,35]. Studies in monkey models revealed that artificially increasing IOP, quantified as the Pressure Insult, subsequently caused an increase in CDR [36,37]. Values of Pressure Insult above 11 mmHg·Days/Day were associated with significant cupping, emphasizing the role of elevated IOP in CDR changes [37]. The study also showed that trabeculectomy to lower Pressure Insult below this threshold was correlated with slowing the rate of CDR progression [37].

In addition to studying IOP and other variables, we examined the relationship between age and LCDR in the POAG population. Although the univariate analysis initially suggested a correlation between older age and LCDR eyes with African ancestry, our multivariate analysis could not confirm this independent association when considering other factors. Thinning of the RNFL with age can influence CDR, as it contributes to structural changes in the optic nerve [38]. Additionally, the development of LCDR can be affected by age-related changes in ocular biomechanics, such as progressive thinning of the outer plexiform layer and increased collagen production leading to tissue stiffness and elevated IOP, as revealed by glaucoma mouse models [39,40,41]. While some studies suggest an age-related decline in neuroretinal rim areas [42], the Rotterdam Study [43] and Singapore Malay Study [31] found no progressive age-related decline, similar to the findings in our study. These nonpathological aging-related changes that affect CDR may contribute to the complexity of interpreting the sole effects of glaucomatous aging-related changes on CDR.

In our study, as in prior studies in different populations, LCDR was also associated with lower MD, higher PSD, and RNFL thinning [35,44]. Visible pores on lamina cribrosa, vessel bayonetting, and nasalization of vessels were also associated with LCDR in our multivariate analysis. Visible pores on the lamina cribrosa may be related to biomechanics-driven lamina cribrosa remodeling and displacement in response to elevated IOP [9,45]. Examinations of glaucomatous laminar pores in both humans and non-human primates have identified the replacement of RGC axons with irregular deposits of extracellular collagen and basement membrane components, clinically observable as optic cupping [46,47,48]. This association between lamina cribrosa pore abnormalities and LCDR highlights the significance of structural changes in the lamina cribrosa in understanding glaucoma progression.

As the optic disc becomes increasingly excavated, the lamina cribrosa may become more susceptible to deformation and displacement, leading to compression and distortion of vessels passing through the optic nerve head (ONH), resulting in vessel bayonetting and nasalization of vessels [49]. A study found that nasalization of the central retinal vessel trunk by 60% or more was significantly associated with glaucoma conversion, indicating progression of disease [49]. In a monkey model of chronic open-angle glaucoma, glaucomatous damage was simulated by increasing IOP. As cupping increased, retinal blood vessels shifted position, particularly moving from far from the ONH edge onto ONH in response to glaucoma damage, possibly further contributing to LCDR [50].

Past studies have shown that while some eyes with LCDRs maintain normal vision, others progress to blindness [17,51,52]. As expected, our multivariate analysis revealed that elevated IOP was strongly predictive of blindness in LCDR eyes. Although individuals with initially normal-range IOPs (between 10 and 20 mmHg) can still develop glaucoma, the Collaborative Normal Tension Glaucoma Trial revealed that lowering IOP by 30% through medical or surgical treatment resulted in reduced rates of glaucoma progression [36]. In African ancestry populations, higher IOP may have more damaging effects on the ONH due to larger optic disc diameters associated with increased vulnerability to pressure-induced deformation [53]. One study found that chronically elevated IOP demonstrated significant axonal dephosphorylation in the retina, ONH, and even the optic chiasm in animal models [36]. This dephosphorylation parallels glaucoma-induced axonal damage, revealing an increase in CDR in all eyes studied and potentially contributing to vision impairment [36]. Additionally, elevated IOP induces cellular stress, leading to retinal astrocyte activation and hypertrophy [54,55]. These hypertrophied astrocytes, observed in experimental rat models, potentially cause hypoxia, which impede the optic axons’ function and contribute to LCDR and optic nerve damage [54].

In our study, LCDR eyes with poor VA were also more likely to have lower (worse) MD, which is associated with faster deterioration rates [56,57]. Paradoxically, lower PSD was associated with blindness. A possible explanation is that, despite the initial gradual elevation of PSD values due to uneven loss of sensitivity in the visual field, advanced glaucoma leads to a general decrease in sensitivity, resulting in lower PSD values overall [58]. Therefore, patients with high IOP and poor visual field measures should be closely monitored for disease progression and considered for more aggressive treatment to prevent visual disability.

Interestingly, previous glaucoma surgery was associated with poor VA in LCDR eyes. However, it should be noted that surgical options are typically only considered in severe glaucoma cases when medical and laser treatments fail to achieve IOP reduction, with the goal of preventing further vision loss [59]. Additionally, surgical interventions can result in complications such as choroidal effusion, shallow anterior chamber, and persistent corneal edema with frequent need for post-operative interventions [60]. Although permanent vision loss, known as the “snuff-out phenomenon”, after trabeculectomy typically occurs in only 2% of patients, various degrees of transient vision loss are reported in up to 50% of patients, and it can take up to two years to recover completely [61].

The genetic analysis for LCDR did not unveil significant associations between the three studied genetic variants and LCDR or blindness among LCDR eyes, indicating that these specific genetic variants might not substantially contribute to the LCDR trait or blindness in POAG. Further studies with larger sample sizes or explorations of other potential risk genes are warranted to better elucidate the genetic underpinnings of VA in LCDR POAG eyes.

This study has some limitations that should be considered. The sample consisted of cross-sectional African American patients from the Philadelphia area, which may limit the generalizability of the findings to the national African American population due to referral bias and geographic isolation. While optic nerve parameters were determined by trained graders with a robust adjudication process, there is a possibility of subjectivity in grading. Moreover, patients with unavailable or low-quality images were excluded from the analysis, introducing potential for selection bias. This study also did not consider the interaction between social, demographic, and biological factors affecting health outcomes. These confounding factors could contribute to elevated risk of poor VA.

In conclusion, in our study, 30% of African ancestry eyes with LCDR progressed to blindness. The strongest risk factors for blindness in African ancestry eyes with LCDR were previous glaucoma surgery, higher IOP, and worse visual field presentation. African ancestry individuals are more likely to experience glaucomatous visual impairment compared to European ancestry individuals, and although management of POAG includes various therapies to reduce IOP, many patients experience substantial limitations in visual function and quality of life despite all treatments. Understanding genetic, demographic, and optic disc phenotypic risk factors for developing poor VA, particularly in patients with LCDRs, is crucial in preventing irreversible blindness.

## Figures and Tables

**Figure 1 genes-14-01809-f001:**
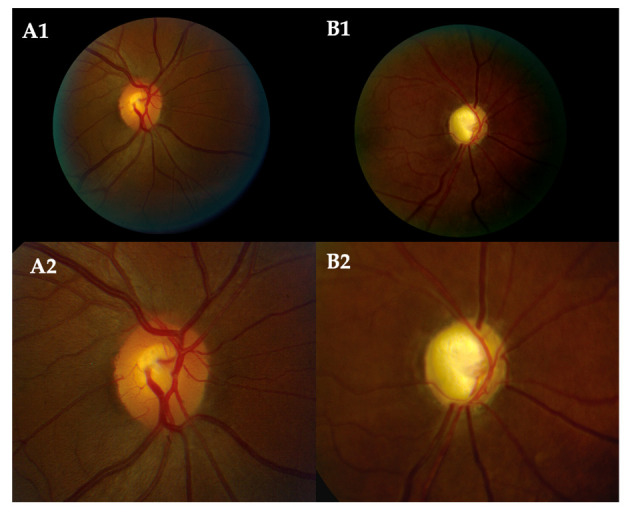
Funduscopic color images of the retinal with examples of a healthy control and LCDR subject. (**A1**) Right eye shows the phenotypic features of the retina of a healthy control subject with a normal optic disc CDR of 0.35. (**A2**) Close-up image of (**A1**) demonstrates the distinction between optic cup and disc use to calculate a CDR of 0.35. (**B1**) Right eye shows the phenotypic features of a retina of a case subject with an optic disc LCDR of 1. (**B2**) Close-up image of the central features of (**B1**) used to calculate a CDR of 1.

**Figure 2 genes-14-01809-f002:**
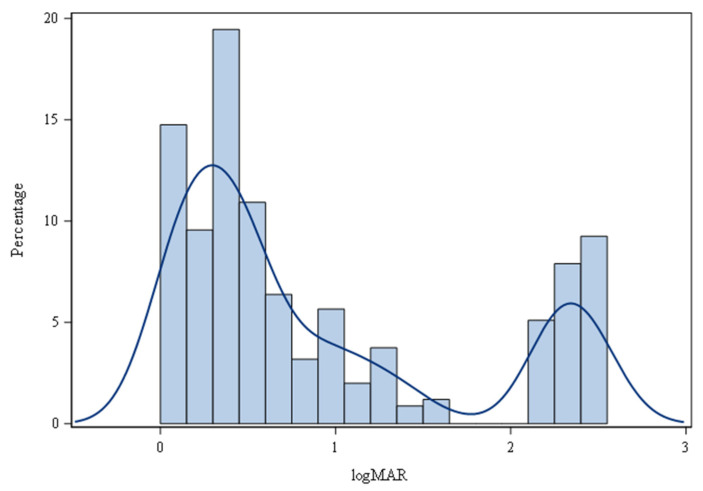
Distribution of VA in LCDR eyes (*n* = 1254). LogMAR = logarithm of the minimum angle of resolution.

**Table 1 genes-14-01809-t001:** Multivariable analysis for association of LCDR ratio with demographic, phenotypic, and optic disc characteristics (*n* = 1890 eyes).

Source	Number of Eyes	LCDR, *n* (%)	Odds Ratio [OR] (95% CI)	*p*-Value
Previous glaucoma surgery				
No	1412	257(18.2%)	Ref	0.002
Yes	478	177 (37.0%)	1.72 (1.23–2.42)	
IOP (per 1 mmHg increase)	1890		1.04 (1.02–1.06)	0.0006
MD (per 1 unit decrease)	1890		1.08 (1.06–1.11)	<0.0001
PSD (per 1 unit increase)	1890		1.06 (1.01–1.12)	0.02
VA (per 1 logMAR unit increase)	1890		1.88 (1.32–2.65)	0.0004
RNFL (per 1 μm decrease)	1890		1.05 (1.04–1.07)	<0.0001
Visible pores in lamina cribrosa				
No	671	84 (12.52%)	Ref	0.004
Yes	1219	350 (28.7%)	1.68 (1.18–2.40)	
Vessel bayonetting				
No	1234	195 (15.8%)	Ref	<0.0001
Yes	656	239 (36.4%)	1.98 (1.44–2.74)	
Nasalization of the vessels				
No	1181	161 (13.6%)	Ref	<0.0001
Yes	709	273 (38.5%)	2.67 (1.92–3.73)	
Shape of cup				
Conical	770	148 (19.2%)	Ref	<0.0001
Cylindrical	863	148 (17.2%)	0.68 (0.48–0.97)	
Bean Pot	248	136 (54.8%)	2.11 (1.35–3.31)	
Others	9	2 (22.2%)	1.55 (0.34–6.96)	

**Table 2 genes-14-01809-t002:** Univariable analysis for association of three genetic variants and LCDR among POAAGG glaucoma cases (*n* = 3242 eyes).

Variants	No. of ALT	LCDR Eyes	Non-LCDR Eyes	OR (95% Cl)	*p*-Value
rs1666698_ALT	0	24 (16%)	124 (84%)	Ref	-
	1	254 (19%)	1074 (81%)	1.22 (0.78, 1.92)	0.383
	2	370 (21%)	1396 (79%)	1.36 (0.88, 2.14)	0.166
rs11824032_ALT	0	355 (19%)	1488 (81%)	Ref	-
	1	262 (21%)	968 (79%)	1.13 (0.95, 1.35)	0.154
	2	31 (18%)	138 (82%)	0.94 (0.63, 1.42)	0.773
rs34957764_ALT	0	585 (20%)	2386 (80%)	Ref	-
	1	63 (23%)	208 (77%)	1.24 (0.93, 1.65)	0.152
	2	0	0	-	-

**Table 3 genes-14-01809-t003:** Multivariate analysis for associations of blindness with phenotypic, demographic, and optic disc characteristics in eyes with LCDR (*n* = 814 eyes).

Source	Number of Eyes	Blind, *n* (%)	Odds Ratio	*p*-Value
IOP (per 1 mmHg increase)	814	384 (59.6%)	1.05 (1.03–1.06)	<0.0001
MD (per 1 unit decrease)	814	384 (59.6%)	1.04 (1.02–1.07)	<0.0001
PSD (per 1 unit increase)	814	384 (59.6%)	0.90 (0.85–0.95)	0.0002
Previous glaucoma surgery				
No	437	75 (17.2%)	Ref	0.0062
Yes	377	109 (28.9%)	1.72 (1.17–2.53)	

**Table 4 genes-14-01809-t004:** Univariable analysis for association of three genetic variants and blind (LCDR eyes with impaired vision) vs. non-blind (LCDR eyes with normal vision) among POAAGG glaucoma cases (*n* = 1050 eyes).

Variants	No. of ALT	Blind	Non-Blind	OR (95% Cl)	*p*-Value
rs1666698_ALT	0	11 (22%)	38 (78%)	Ref	-
	1	97 (23%)	316 (77%)	1.06 (0.53, 2.13)	0.870
	2	153 (26%)	435 (74%)	1.22 (0.62, 2.40)	0.573
rs11824032_ALT	0	148 (25%)	437 (75%)	Ref	-
	1	96 (23%)	313 (77%)	0.91 (0.69, 1.19)	0.478
	2	17 (30%)	39 (70%)	1.29 (0.72, 2.31)	0.399
rs34957764_ALT	0	238 (25%)	710 (75%)	Ref	-
	1	23 (23%)	79 (77%)	0.87 (0.54, 1.39)	0.557
	2	0	0	-	-

## Data Availability

The data presented in this study are available on request from the corresponding author. The data are not publicly available due to their containing information that could compromise the privacy of research participants.

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
