# Peer review of "Factors Associated with Large Cup-to-Disc Ratio and Blindness in the Primary Open-Angle African American Glaucoma Genetics (POAAGG) Study"

_genes, 2023, doi:10.3390/genes14091809_

Round 1
Reviewer 1 Report
The present study described the factors associated with large cup-to-disc ratio and blindness and the fingding is helpful for the treatment of glucoma. I have the following major points.
1. The typical retinal fundus photograph of the patients with large cup-to-disc and those of the healthy control might be provided.
2. The selected factors associated with large cup-to-disc ratio and blindness might be confirmed in the patients and animal models, or they might be discussed in the discussion sections.
The language is ok.
Reviewer 2 Report
The present study is trying to address the risk factors associated with the development of LCDR and blindness in African ancestry with POAG. The study's key findings provide insights into the relationship between high LCDR values, retinal ganglion cell loss, and visual impairment in individuals of African ancestry with POAG. This could potentially lead to a better understanding of factors that influence the progression of POAG and its impact on vision within this specific population. Even though the authors are trying to address a very significant question in POAG, a few comments need to be addressed to enhance the manuscript further.
Major comments:
1. Is there any data available for comparing LCDR in individuals of African ancestry with those of Asian ancestry, rather than solely comparing with individuals of European ancestry? The authors have mentioned that existing studies in European and Asian ancestry populations have found that risk factors for LCDR include higher IOP, lamina cribrosa depth, age, and gender. These points need to be further addressed within the discussion in comparison to other ancestries.
2. In addition, age as a risk factor in correlation to African American race should also be highlighted and how that would impact LCDR.
Minor comments:
1. In the introduction, lines 52-54, the authors have mentioned that there are “various studies” that address LCDR and visual field loss in POAG patients. However, the authors have quoted only a single reference for it.
2. In the introduction, lines 56-57, the authors cited previous literature for evaluating LCDR with normal vision and blindness. However, the authors have not referenced the outcomes of that study.
3. Since there are vertical CDR and LCDR, it is better to specify which type of CDR the authors mentioned in the introduction, line 54.
4. The authors can use LCDR instead of larger CDR throughout the text after its initial usage to maintain consistency (E.g., in lines 52 and 209).
5. The manuscript needs to be proof-read for uniformity of font size.
